# Effects of Energy Drink Acute Assumption in Gastrointestinal Tract of Rats

**DOI:** 10.3390/nu14091928

**Published:** 2022-05-04

**Authors:** Milena Nasi, Anna De Gaetano, Gianluca Carnevale, Laura Bertoni, Valentina Selleri, Giada Zanini, Alessandra Pisciotta, Stefania Caramaschi, Luca Reggiani Bonetti, Alberto Farinetti, Andrea Cossarizza, Marcello Pinti, Antonio Manenti, Anna Vittoria Mattioli

**Affiliations:** 1Department of Surgery, Medicine, Dentistry and Morphological Sciences, University of Modena and Reggio Emilia, 41121 Modena, Italy; milena.nasi@unimore.it (M.N.); gianluca.carnevale@unimore.it (G.C.); laura.bertoni@unimore.it (L.B.); alessandra.pisciotta@unimore.it (A.P.); antonio.manenti@unimore.it (A.M.); annavittoria.mattioli@unimore.it (A.V.M.); 2National Institute for Cardiovascular Research (INRC), 40126 Bologna, Italy; anna.degaetano@unimore.it (A.D.G.); valentina.selleri@unimore.it (V.S.); 3Department of Life Sciences, University of Modena and Reggio Emilia, 41121 Modena, Italy; giada.zanini@unimore.it; 4Department of Medical and Surgical Sciences for Children and Adults, University of Modena and Reggio Emilia, 41121 Modena, Italy; stefania.caramaschi@unimore.it (S.C.); luca.reggianibonetti@unimore.it (L.R.B.); alberto.farinetti@unimore.it (A.F.); andrea.cossarizza@unimore.it (A.C.)

**Keywords:** energy drinks, caffeine, eosinophils

## Abstract

Energy drinks (EDs) are non-alcoholic beverages containing high amounts of caffeine and other psychoactive substances. EDs also contain herbal extract whose concentration is usually unknown. EDs can have several adverse effects on different organs and systems, but their effects on the gastrointestinal (GI) tract have been poorly investigated. To determine the acute effects of EDs on the GI tract, we administered EDs, coffee, soda cola, or water to Sprague–Dawley rats (n = 7 per group, randomly assigned) for up to five days, and analyzed the histopathological changes in the GI tract. Data were compared among groups by Kruskal–Wallis or Mann–Whitney tests. We found that, while EDs did not cause any evident acute lesion to the GI tract, they triggered eosinophilic infiltration in the intestinal mucosa; treatment with caffeine alone at the same doses found in EDs leads to the same effects, suggesting that it is caffeine and not other substances present in the EDs that causes this infiltration. The interruption of caffeine administration leads to the complete resolution of eosinophilic infiltration. As no systemic changes in pro-inflammatory or immunomodulating molecules were observed, our data suggest that caffeine present in ED can cause a local, transient inflammatory status that recruits eosinophils.

## 1. Introduction

Energy drinks (EDs) are non-alcoholic beverages containing a high amount of caffeine and several other psychoactive substances, including the amino acid taurine, the glucose derivative glucuronolactone, and herbal extracts such as ginseng and guaranà (another source of caffeine, with caffeine-like effects), often present in uncertain concentrations [1]. EDs were placed on the market in the 1960s in Europe and Asia and they have spread worldwide since the end of the last century [2]. The short- and long-term health effects of EDs consumption on the cardiovascular and central nervous system have been intensively studied.. EDs have been shown to cause several adverse effects in humans, and particularly in young people, including high blood pressure, changes in corrected QT interval (QTc), serious cardiovascular events, kidney disorders, metabolic adverse effects, poor sleep, seizures, and neuropsychiatric adverse effects [3,4,5,6]. Less attention has been paid to the impact on the gastrointestinal (GI) tract and, indirectly, on the immune system. Most of the data concerning GI effects are inferred by studies on humans with caffeine, whose effects are contradictory [7]. It has been demonstrated that esophageal reflux can occur in EDs consumers, even if it is not associated with dyspepsia symptoms [8,9]. Furthermore, caffeine has a significant effect on colorectal activity [10] and, at least in some cases, can lead to GI upset [11]. Caffeine also impacts glucose absorption, as higher levels of glucose and insulin are observed when sweetened caffeinated—but not decaffeinated—beverages are assumed [12]. However, it must be noted that several of the GI effects of coffee are not attributable to caffeine, as they are similar to those of decaffeinated coffee, or are not totally lost after decaffeination. A study of humans showed that the infusion of caffeine at doses present in a cup of coffee resulted in a net GI fluid secretion, with no effects on small bowel transit [13]. Other components of coffee can counteract this effect, as perfusion with coffee does not determine a net fluid secretion [14]. Finally, coffee stimulates the release of cholecystokinin, leading to gallbladder contraction which may exacerbate symptoms in patients with symptomatic gallstones [7].

Although multiple effects of caffeine or caffeine-containing beverages have been observed on the GI tract, little attention has been paid to the possible contribution of these beverages to inflammation or the activation of immune cells. Recently, a study reported that EDs consumption can exert an anti-inflammatory effect in vitro, by reducing the release of Interferon (IFN)-γ by endothelial cells, the secretion of Interleukin (IL)-6 and Tumor necrosis factor (TNF)-α by cell lines of monocytic origin, and can reduce experimental colitis in rats, suggesting that EDs can have an effect on inflammation and immune response [15]. In this study, we aimed at comparing the acute effects of EDs on the GI tract with other caffeine-containing beverages such as soda cola and coffee and determining if their effects can be attributed to caffeine. We provide some lines of evidence of a pro-inflammatory, eosinophilic effect of acute assumption of EDs on the GI tract in an animal model. Furthermore, we show that these effects are indistinguishable from those from coffee containing comparable levels of caffeine, suggesting that this effect is not related to other ingredients present in EDs besides caffeine. 

## 2. Materials and Methods

### 2.1. Animals and Dietary Treatment

Two sets of experiments were performed in vivo on rats. In the first set of experiments, a total of twenty-eight male adult Sprague–Dawley rats, not genetically related, were used. Sprague–Dawley rats were chosen because of their human-like digestive physiology. The rats were fed a dry standard laboratory diet, and their weights ranged from 230 to 250 g. They were housed in cages for a 12 h day/night cycle with controlled humidity and temperature. The animals were randomly divided into 4 groups, each composed of 7 animals receiving different sweetened beverages containing caffeine and caffeine-like substances. Group A: control group, receiving water with a known composition. Group B: commercial ED for 5 days. Group C: commercial soda cola for 5 days. Group D: commercial sweetened coffee for 5 days. The amount of each beverage was calculated considering the intake of caffeine ingested by an adult man of 70 kg of body weight consuming two cans/day of ED or soda cola, or 4/5 cups of espresso. The animals were sacrificed at the end of the treatment using the hypercapnia technique. 

In the second set of experiments, adult male rats, weighing 230–250 g and fed with a standard laboratory diet, were randomly divided into 2 groups, each one consisting of 2 subgroups (N = 5), one supplemented with water (control, subgroup A), the other with coffee (subgroup B). Both the groups received the treatment for 5 days followed by 5 days of observation before the sacrifice for the first group (group 1) and 10 days for the second (group 2). The amount of coffee was calculated in order to provide the same amount of caffeine taken by the rats fed with ED in the first experiment. 

All of the animals received human care in compliance with the European rules of experiments in animals (European Directive 2010/63/EU). The study was conducted according to the guidelines of the Declaration of Helsinki. The animal protocols were approved by the the Ethics Committee of Italian Ministry of Health, according to Italian law protecting animals used for scientific purposes (authorization n° 544/2016 PR, released on 30 May 2016 and next update released on 11 September 2017 from Italian Ministry of Health). 

### 2.2. Examination and Isolation of Organs and Tissue

At autopsy, the organs connected with beverage contact, absorption, and metabolism, namely the entire GI tract, including the esophagus, stomach, duodenum, small intestine and colon, liver and pancreas, the immunocompetent organs, such as the spleen, thymus, and retro-peritoneal lymph nodes, and the thoracic organs, i.e., heart and lungs, were macro- and microscopically examined, and seriated specimens were taken from the stomach and intestine.

### 2.3. Histological Stains and Immunohistochemistry

Besides hematoxylin/eosin (H&E), Pagoda red (Dylon International LTD, England) staining was performed to quantify eosinophils, as described by Trani et al. [16]. A portion of each sample of stomach and intestine was washed in Phosphate-Buffered Saline (PBS), fixed in 4% paraformaldehyde (PFA) in PBS, then rinsed with PBS, dehydrated with graded ethanol, and cleared and embedded in paraffin. Five µm thick serial cross sections of the specimens from each experimental group were obtained. Routine H&E staining was performed in order to analyze the morphological details of the tissues [17], whereas the presence of inflammatory cells was investigated using immunohistochemical analyses by incubating the samples at +4 °C overnight with the following primary antibodies: mouse anti-rat IL-4 (LSbio, Seattle, WA, USA), goat anti-rat IL-5, goat anti-rat IL-6 (R&D systems, Minneapolis, MN, USA), and rabbit anti-IL-33 (Novus Bio, St. Louis, MO, USA), all diluted 1:50 in PBS containing 1% bovine serum albumin (BSA). Subsequently, the samples underwent incubation with anti-mouse, anti-goat, and anti-rabbit horseradish peroxidase (HRP)-linked secondary antibodies (Invitrogen, Waltham, MA, USA; all diluted 1:100 in PBS) and were finally revealed by a 3,3’-diaminobenzidine (DAB)-based kit (Sigma Aldrich, St. Louis, MO, USA).

### 2.4. Plasmatic Soluble Factors Analysis

Plasma was collected from whole blood samples at the end of the treatment. Soluble factors, markers of tissue damage and cytokines, were analyzed in plasma samples using a magnetic-bead-based multiplex assay for the Luminex^®^ platform (RandD System, Minneapolis, MN, USA), using a “Luminex Performance Human High Sensitivity Cytokine Magnetic Panel A”. The following soluble factors were quantified: IL-1β, IL-2, IL-4, IL-6, IL-10, IL-33, TNF-α, IFN-γ, TIMP metallopeptidase inhibitor (TIMP) -1, Vascular endothelial growth factor (VEGF), ICAM, and L-selectin. 

### 2.5. Analysis of Circulating mtDNA in Plasma

Droplet digital PCR was used to measure the amount of circulating mitochondrial DNA (mtDNA). Total DNA was extracted from rat plasma samples using a QIAmp DNA Minikit, Qiagen (Alameda, CA, USA), following the manufacturer’s instructions. An amount of 1 µL of total DNA was added to a 20 µL final volume mixture composed of: 10 µL of 2× ddPCR Supermix for Probes, 1 µL of ddPCR assay for ND2 (UniqueAssayID, dHsaCPE5043508), 1 µL of ddPCR assay for EIF2C1 (UniqueAssayID, dHsaCP2500349), and 7 µL of nuclease-free water (all reagents from Bio-Rad, Hercules, CA, USA). The droplet generations and readings were performed using the Bio-Rad QX200 ddPCR droplet system (Bio-Rad laboratories) [18,19]. Circulating mtDNA content was expressed as the number of copies per mL of plasma.

### 2.6. Statistical Analysis

Statistical analyses were performed using Prism 8.0 (GraphPad, San Diego, CA, USA). The results are expressed as the mean ± Standard Deviation (SD). Non-parametric Kruskal–Wallis or Mann–Whitney tests were used to compare quantitative variables. A *p* value < 0.05 was considered significant.

## 3. Results

### 3.1. Effects of the Dietary Treatment on Organ Macroscopic Phenotype and Metabolic Parameters

In order to determine the effects of EDs on the GI tract, we administered EDs, soda cola, or sweetened coffee to Sprague–Dawley rats for five days. The rats tolerated treatments without general or digestive complications. However, a weight gain of 7 ± 4.5% and 5 ± 3% (mean ± SD) was observed in animals receiving ED and soda cola, respectively. Furthermore, the rats supplemented with ED also showed increased excitability and difficult handling.

The macroscopic examination of abdominal and thoracic organs did not disclose evident lesions. All of the common metabolic parameters tested just before the sacrifice were always in a normal range, as were the blood counts, including leukocytes and eosinophils (not shown).

Histological examination of the GI tract excluded secondary lesions ingroup A of control animals supplemented with water, whereas an eosinophilic infiltration was observed in the mucosa of the stomach and duodenum of the animals supplemented with coffee, EDs, and soda cola (Figure 1).

This pathology involved superficial epithelial cells and in full-thickness mucosa, submucosa until the lamina propria, but not the muscularis and serosa layers. Eosinophils showed a mature feature, with a normally sized nucleus, often ring-shaped, and a cytoplasm densely packed with granules. The esophagus appeared free from any lesion (not shown), while evident eosinophilia also involved the spleen, with a perifollicular infiltration not distorting its basal architecture (Appendix A). We next quantified the number of infiltrating eosinophils (Figure 1). In the stomach, the highest level of eosinophilic infiltration was visible in rats supplemented with EDs, while a moderate, not significant, increase appeared in soda cola and coffee (Figure 1). In the intestine, it appeared at the highest levels in rats treated with coffee, EDs showed a moderate increase, while soda cola did not evidence a significant eosinophilic infiltration. To determine which could be the cause of eosinophilic infiltration, we measured a series of parameters related to inflammation and to Th1/Th2 response, namely the plasmatic concentration of pro-inflammatory and Th1/Th2 cytokines, along with circulating mtDNA as an indirect marker of cell damage and innate inflammation. 

The levels of pro-inflammatory cytokines did not reveal any condition of systemic inflammation. TNF-α and IL-6 were below the limits of detection in all conditions tested, while IL-1 β displayed a high degree of variability but was always in the range of physiological concentrations (Figure 2).

Concerning Th1 response, all conditions tested showed plasma concentrations of IFN-γ similar to those observed in animals receiving water. IL-4 and IL-13, two typical Th2-response cytokines, were undetectable in all conditions analyzed. Taken together, these results suggested that EDs, similarly to other caffeine-containing beverages, did not cause any systemic inflammation or Th2 immune polarization, despite the activation of a local eosinophilic response. 

### 3.2. Caffeine Triggered the Eosinophilic Infiltrate in Intestinal Mucosa

As the effects observed in the GI tract were present in rats treated with coffee, cola, and ED, we wondered if it could be attributed to caffeine. Thus, to rats, we administered a coffee solution containing caffeine at the same concentration taken by rats supplemented by ED in the previous experiment for up to 5 days, followed by a recovery time of 5 or 10 days without caffeine supplementation (Figure 3). The five days of caffeine administration led to a uniform infiltration of eosinophils in the mucosa and in the sub-mucosa, which did not involve the epithelia of the GI glands, as well as blood and lymphatic capillaries (Figure 3A). In the stomach, a mild and transient hyperemia was detected, lasting only 5 days. Eosinophil infiltration reached its maximal levels after 5 days of caffeine supplementation, persisted for 5 days (Group 1, subgroup B), and then decreased progressively until complete disappearance after 10 days (Group 2, subgroup B; Figure 3B). Subgroup A, used as controls, showed no eosinophilic infiltration. The eosinophils involved in this process had a typical murine appearance, characterized by a big, ring-shaped nucleus and multiple eosinophilic granules in the cytoplasm. Contemporary splenic eosinophilic colonization was observed after 5 days from the interruption of caffeine supplementation, involving the peri-sinusoidal spaces and not the splenic follicles. It disappeared after 10 days, at the same time as the resolution of the original eosinophilic infiltration in the mucosa of the GI tract. Moreover, the resolution of the eosinophilic infiltration in these tissues occurred without any residual histological abnormality, related to necrosis or the recruitment of inflammatory cells.

## 4. Discussion

In this study, we described the effects of the assumption of EDs on the GI tract. 

The analysis of rats from a phenotypic point of view showed weight gain and increased excitability. The same effects have been largely described in humans assuming EDs [20] and are attributed to the high content of sugar and caffeine in EDs.

Thus far, the only study describing the effect of EDs on inflammation limited its analysis to the GI tract [15]. Our study provided some clues on this point, through the analysis of metabolic and inflammatory factors related to inflammation, such as mtDNA, or to Th1/Th2 response. Considering that IL-4 and IL-13 are cytokines of the eosinophil/Th2 axis, TIMP-1 is a player in preserving tissue integrity [21], and the circulating form of ICAM-1 is associated with inflammation and endothelial damage [22], these observations suggest that the consumption of EDs does not dramatically affect peripheral inflammatory response. This is also confirmed at organ level, as shown by the examination made during the autopsy in which no macroscopical abnormality was observed.

The microscope analysis revealed eosinophilic infiltration in the GI tract and spleen. Eosinophils are proinflammatory leukocytes specialized in parasitic infections and tissue damage, also involved in different cell processes associated with body homeostasis and cancer immunology [23]. The observation that the eosinophilic infiltration of the GI mucosa occurred without primary involvement of the hematopoietic or immunocompetent organs suggested that eosinophils derived from progenitor cells homed in the gastrointestinal mucosa, inside the lamina propria. Indeed, eosinophils within the GI exist physiologically in a steady state promoting tissue homeostasis as well as an immune and inflammatory response [24]. 

Eosinophilic infiltration could be triggered by direct contact between an extraneous, but not tissue toxic, substance, the caffeine, and the gastrointestinal mucosa, which responded with a local allergic-type reaction and recruitment of eosinophils. The splenic eosinophilic infiltration, observed after 5 days from the interruption of caffeine supplementation, confirmed the exuberant production of these cells that found a compatible refugee-organ in the spleen, so confirming its good receptivity for myeloid-derived cells. This passive colonization was not followed by any secondary eosinophilic recirculation or splenic activation and resolved completely at the same time as the original gastrointestinal mucosal infiltration. The local gastrointestinal eosinophilopoiesis typically agreed with the absence of general mediators such as immunoglobulins and cytokines [25]. Moreover, the complete resolution of the gastrointestinal eosinophilia coincided with the normal eosinophil life span, likely as a result of a physiological process of apoptosis of eosinophils [26]. All these features suggested that the eosinophilia was not an inflammatory process, but rather a chemically induced reaction to an exogenous substance, likely caffeine. This is in agreement with the fact that caffeine is rapidly absorbed in the stomach and small intestine [27]. The absence of any eosinophilic infiltration of the esophageal mucosa could be due to the short time of contact with the caffeine, and the protective effect of its lining epithelium [28].

In humans, eosinophilic stomach and small intestine infiltration is characterized by eosinophilic gastroenteritis (EGE). EGEs are eosinophil-associated diseases (EADs), rare pathologies characterized by an increase in the circulation or in the tissue number of eosinophils [24,29]. When the eosinophilic infiltrate is present in GI tracts, the diseases are named eosinophilic gastrointestinal disorders (EGIDs). Eosinophilic infiltration characterizing EGIDs can affect different layers of the bowel wall such as mucosal, muscular, or serosal. Generally, patients affected by EGE show excess eosinophils in the GI tract and high levels of IgE, suggesting the presence of an allergic component and an implication of a mechanism involving Th2, even if the origination and the development of this disease is not well understood [30,31,32]. Eosinophilic esophagitis is the most studied type of EGID with a standardized diagnosis and therapy, whereas the data on the other types of EGIDs are limited even if eosinophilic gastritis, gastroenteritis, and colitis are increasing in their prevalence in the last decade, making further research necessary [33]. Acute caffeine treatment in our study triggered eosinophilic infiltration in the rats comparable to that one observed in EGE, making this treatment a good model to study acute eosinophilia, and laying the foundation for further studies necessary to highlight the origin and development of chronic eosinophilic infiltration typical of EGIDs. 

Our results can have relevant implications when translated into clinics, particularly as far as the potential negative effects of EDs on human health, and especially on young people, are concerned. The intake of beverages with a high caffeine content can cause acute effects (i.e., the development of arrhythmias) and chronic effects on which little information is available. The synergy of the high caffeine dosage with other components of the drink such as taurine can enhance these negative effects on human health. Specifically, the effects detected on the GI tract with an enhancement of the pro-inflammatory (and/or immunological) action can facilitate the onset of diseases. EDs also contain an important quantity of sugars. The consumption of sugar-sweetened beverages predicts poor health outcomes in the aging population, including cardiovascular disease, diabetes, and cancer [34,35]. Evidence further supports a link between sugar-sweetened-beverages-triggered pathological processes and biologic factors of aging, including inflammaging, oxidative stress, and alterations in intestinal microbiota [34,36]. The GI tract is crucial in maintaining the homeostasis of several physiological processes, including immunological tolerance to foods and commensal microorganisms (gut microbiota) [37]. Thus, modifications in the microbiome structure can be a consequence of, besides reduced intestinal functionality, the use of medications and recurrent infections or a dietary pattern that includes beverages such as EDs. From this perspective, another aspect that should be analyzed is the modification of the microbiota following the chronic intake of EDs. 

However, our study has some limitations. First, we did not have the possibility to dissect the effects of the single molecules of herbal extracts normally present in EDs, as their actual content and concentration are not disclosed. Second, our study is limited to short-term, acute effects, and did not explore the effects of long-term treatment with EDs. Finally, our study does not provide information concerning the functional consequences of eosinophilic infiltration on the GI tract. Thus, further studies are needed to determine if the eosinophilic infiltration we observed could, when prolonged over time, contribute to EGIDs. 

## 5. Conclusions

In summary, these results show that, despite eosinophilia, the acute assumption of EDs does not have deleterious consequences when compared with coffee. Moreover, our data show that acute caffeine treatment in rats could be a good model to study the origin and the development of acute eosinophilia in the stomach and intestine. 

## Figures and Tables

**Figure 1 nutrients-14-01928-f001:**
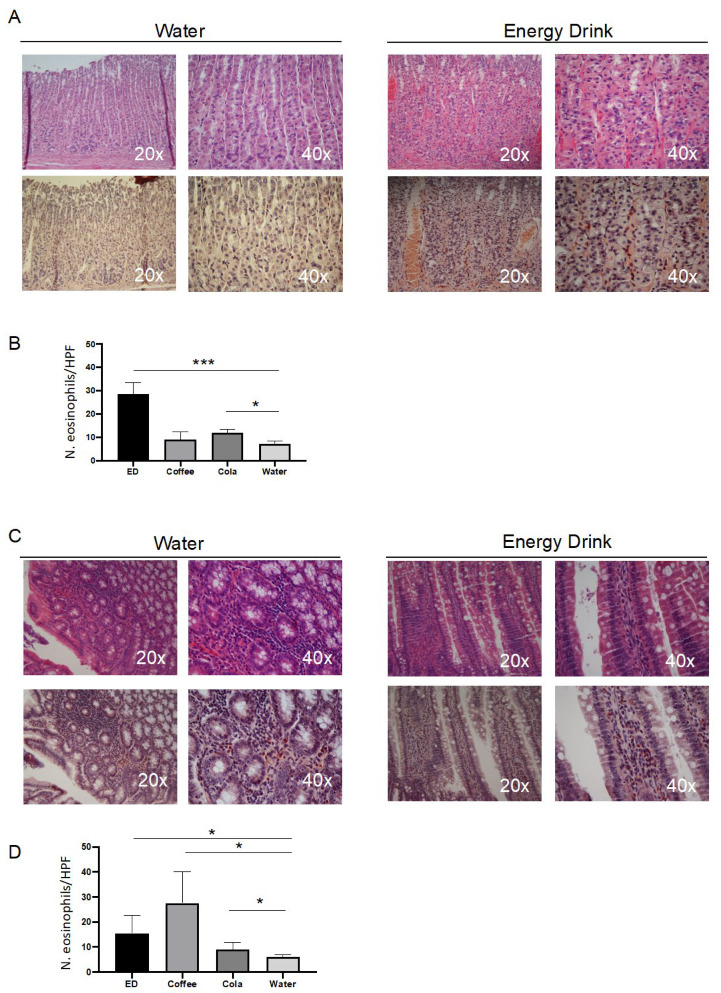
Energy drinks (ED) and coffee trigger eosinophilic infiltration in the mucosa of stomach and intestine in rats. (**A**). Representative histological staining of stomach sections from after 5 days with the indicated treatment. Upper panels: hematoxylin/eosin (H&E) staining; lower panels: Pagoda red staining. (**B**). Histogram showing the number of eosinophils per high-powered field (HPF) after five days of the indicated treatment. Data are mean ± SD of ten different counts. (**C**). Representative histological staining of small intestine sections from after 5 days with the indicated treatment. Upper panels: H&E staining; lower panels: Pagoda red staining. (**D**). Histogram showing the count of eosinophils per HPF after five days of the indicated treatment. Data are median ± interquartile range of ten different counts. * *p* < 0.05; *** *p* < 0.001.

**Figure 2 nutrients-14-01928-f002:**
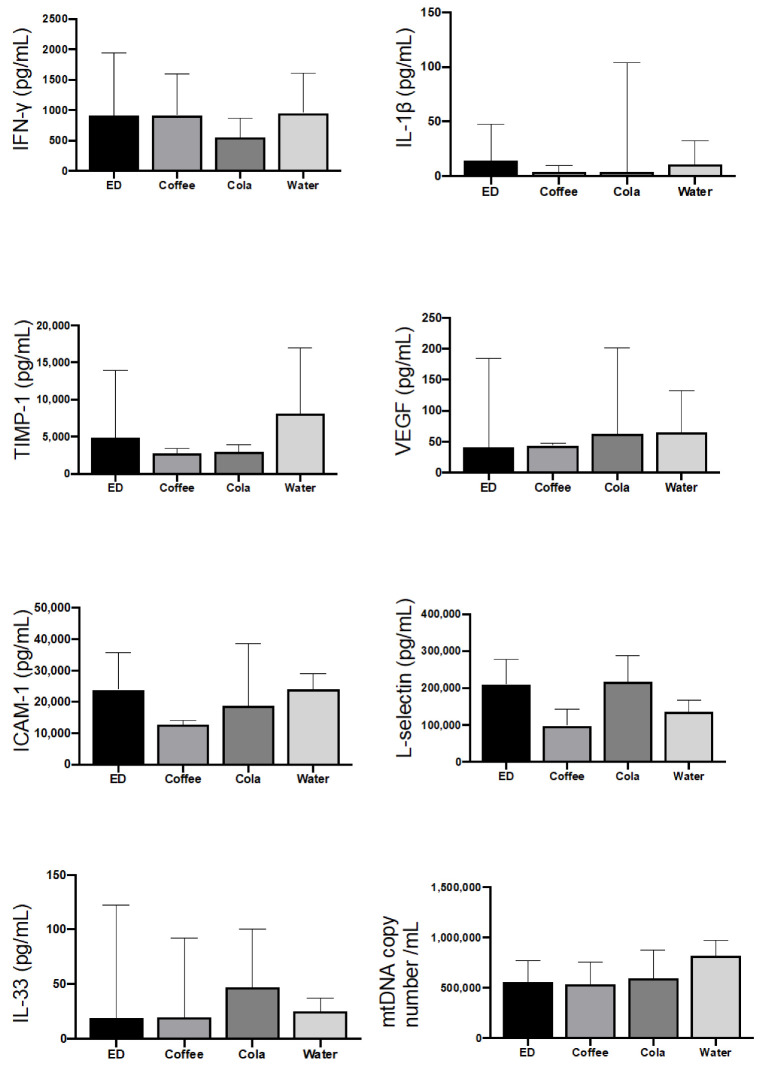
Caffeine does not induce systemic inflammation, despite eosinophilic reaction in the GI tract. Plasma levels of the indicated immunomodulatory molecules, as measured in rats after 5 days of caffeine administration. Concentrations are expressed as pg/mL, with the exception of mtDNA, which is expressed as number of copies per mL. Data are median ± interquartile range of seven samples, each in duplicate. ED = Energy Drink.

**Figure 3 nutrients-14-01928-f003:**
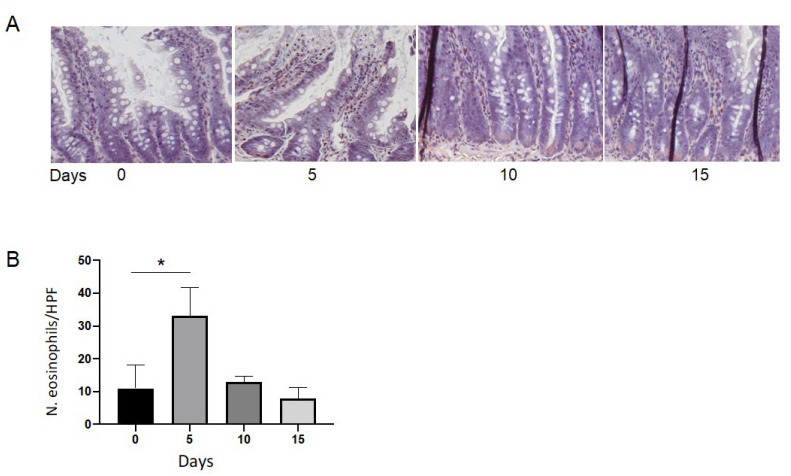
Caffeine triggers eosinophilic infiltration in the mucosa of intestine in rats. Rats assumed a beverage containing caffeine for five days, followed by ten days without assumption. (**A**). Representative Pagoda red staining of intestine sections before caffeine assumption (0), after 5 days of treatment (5), after five more days without caffeine assumption (10) and after ten days without caffeine assumption (15). (**B**). Histogram showing the number of eosinophils per HPF at the same time points. Data are median ± interquartile range of five independent counts on five animals. * = *p* < 0.05.

## Data Availability

The data presented in this study are available on request from the corresponding author.

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
