# Peer review of "Effects of Energy Drink Acute Assumption in Gastrointestinal Tract of Rats"

_nutrients, 2022, doi:10.3390/nu14091928_

Round 1

Reviewer 1 Report

Your article evaluates the effects of energy caffeinated drinks on GI of rats. The article respects the general principles of a research article. The methods are aquartely described, the results are cleraly presented. The discussion chapter must be improved. please describe the potenatial clinical relevance of your results. please analyse the potential impact of sweetened beverages that are enriched with caffeine on the gastrointestinal tract of humans. Is the sugar in these drinks responsable for a pro-inflammatory effect at the level of the GI tract? what role does caffeine have? Please check: A.R. Popa, C.M. Vesa, D. Uivarosan, C.M. Jurca, G. Isvoranu, B. Socea, A.M.A. Stanescu, M.A. Iancu, I. Scarneciu, D.C. Zaha, Cross Sectional Study Regarding the Association Between Sweetened Beverages Intake, Fast-food Products, Body Mass Index, Fasting Blood Glucose and Blood Pressure in the Young Adults from North-western Romania. Revista de Chimie, vol. 70, no. 1, pp. 156-160, 2019. . https://doi.org/10.37358/RC.19.1.6872

Author Response

We thank the reviewer for her/his precious comments, which give us the possibility to better clarify some aspects of our study, and for pointing our attention to the study by Popa et al., which has been mentioned and cited in the bibliography. We added this paragraph in the discussion, where the importance of sugar as a possible cause of inflammation, and the clinical implications of our study are discussed:

“Our results can have relevant implications when translated into clinics, particularly as far as the potential negative effects of EDs on human health, and especially on young people, is concerned. The intake of beverages with a high caffeine content can cause acute effects (i.e. the development of arrhythmias) and chronic effects on which little information is available. The synergy of the high caffeine dosage with other components of the drink such as taurine can enhance these negative effects on human health. Specifically, the effects detected on the GI tract with the enhancement of the pro-inflammatory (and/or immunological) action can facilitate the onset of diseases.  EDs also contain an important quantity of sugars. Consumption of sugar-sweetened beverages predicts poor health outcomes in the aging population, including cardiovascular disease, diabetes, and cancer [34,35]. Evidence further supports a link between pathological processes triggered by sugar-sweetened beverages and inflammaging, alterations in intestinal microbiota, oxidative stress and inflammaging [34,36]. The GI tract is crucial in maintaining the homeostasis of several physiological processes, including immunological tolerance to foods and commensal microorganisms (gut microbiota) [36]. Thus, modifications in the microbiome structure can be a consequence of, besides reduced intestinal functionality, use of medications and recurrent infections, a dietary pattern that include beverages such as EDs. In this perspective, another aspect that should be analyzed is the modification of the microbiota following the chronic intake of EDs.”

Reviewer 2 Report

The manuscript deals with a study the effect of energy drinks acute assumption in gastrointestinal tract of rats. The findings of in vivo performed experiments revealed that energy drinks have undesirable influence on rats gastrointestinal tract. They have a significant impact on GI tract and splenic eosinophilic infiltration. Moreover, the weight (mass) gain has been observed due to the high content of sugar and caffeine in the energy drinks.

This research is needed as the use of energy drinks grown exponentially. However, it is well known from the literature data that regular use of energy drinks has an influence on human health. The numerous studies have shown that excess consumption of energy drinks may result in health-consequences similar to those from excess exposure to caffeine. Moreover, energy drinks may increase the rate of alcohol-related injury and may serve as a gateway to other forms of drug dependence, especially in the case of children and adolescents who are not habitual caffeine users.

Before acceptance, the authors should defined all abbreviations and acronyms used in the manuscript. For example, SD (page 3), H&E (page 5), and others.

Author Response

We thank the reviewer for appreciating our study. As requested, we defined all the abbreviations used in the text.

Reviewer 3 Report

Nutrients-1665446: Effects of Energy drink acute assumption in gastrointestinal tract of rats.

Dear authors,

This is a concrete and clear work. Thank you.

Below are my comments:

Abstract:

  1. Always use EDs.
  2. “with chose caused by coffee and soda cola” error?
  3. Indicate de number of rats. How did you decide which ones to give soda to and which ones to give GI? Explain if there were different groups.
  4. What did you use to assess mucosa? What statistical analysis did you use to estimate the differences between drinks?

Introduction:

  1. Throughout the introduction you talk about different study results, but are they in animals or in humans? It would be advisable to specify.
  2. In the last paragraph, in addition to what "this article adds" you should add your objective.

Methods

  1. Did you choose in any particular way which brand of which drink to give to the animals?
  2. Did you do any assessment of the animals before you started giving the drinks? To be able to compare before and after.
  3. Statistical analysis: why do you describe numerical variables as mean and standard deviation if you have used non-parametric tests? You should then describe them as median and interquartile range.
  4. It is highly recommended to add a table, even if it is table 1 of any article, a descriptive table. The results are too extensive considering that the analysis carried out is descriptive. I recommend you to reduce and specify much more the information you present. To do this, you should be very specific in your objectives and focus on just one.

Discussion

  1. More references need to be included between 24 and 25.
  2. It is very important to add a paragraph on the limitations of this study.

Thank you.

Author Response

This is a concrete and clear work. Thank you.

ANSWER: We thank the reviewer for appreciating our study.

Abstract:

Always use EDs.

ANSWER: we make the use of “EDs” term consistent.

“with chose caused by coffee and soda cola” error?

ANSWER: Yes, it was an error, that we fixed.

Indicate de number of rats. How did you decide which ones to give soda to and which ones to give GI? Explain if there were different groups.

ANSWER: We added the number of animals in the abstract, and we underlined that groups were randomly formed.

What did you use to assess mucosa? What statistical analysis did you use to estimate the differences between drinks?

ANSWER: We added statistical tests used in the abstract.

Introduction:

Throughout the introduction you talk about different study results, but are they in animals or in humans? It would be advisable to specify.

ANSWER:  Most of the studies cited have been performed in humans. We specified it when it was not already indicated, or obvious.

In the last paragraph, in addition to what "this article adds" you should add your objective.

ANSWER: We added the aim of the study in the Introduction, as requested.

Methods

Did you choose in any particular way which brand of which drink to give to the animals?

ANSWER: We have chosen the most popular and widespread brands of EDs and Cola, which are, among teenagers and young people,  the ED and Cola par excellence.

Did you do any assessment of the animals before you started giving the drinks? To be able to compare before and after.

ANSWER:  In the first set of experiments, we did not do any assessment other than weight, as we compared animals with different treatments, and we need to sacrifice in order to get results. The second set of experiments was longitudinal, so we had different groups of animals and in a group, we performed histochemical analyses before treatment, while the other four groups were analyzed at different times after starting treatment, as stated in the legend of the figure 3.

Statistical analysis: why do you describe numerical variables as mean and standard deviation if you have used non-parametric tests? You should then describe them as median and interquartile range.

ANSWER:  As requested, we modified the way data are presented, showing median and interquartile range.  The figures have been updated accordingly.

It is highly recommended to add a table, even if it is table 1 of any article, a descriptive table. The results are too extensive considering that the analysis carried out is descriptive. I recommend you to reduce and specify much more the information you present. To do this, you should be very specific in your objectives and focus on just one.

ANSWER: We thank the reviewer for her/his comment. We respectfully disagree with the need a Table 1 with descriptive data, as it is not usually added in studies concerning rats, as their main features are typical of the strain used, and extremely homogenous.

We agree with the need of being more synthetic in the Results section, and we slightly shortened it.

Discussion

More references need to be included between 24 and 25.

ANSWERS: We cited some of the most relevant studies to support observations made in the Discussion, between ref 24 and 25:

  1. Yan, B.M.; Shaffer, E.A. Primary eosinophilic disorders of the gastrointestinal tract. Gut 2009, 58, 721-732, doi:10.1136/gut.2008.165894.
  2. Rothenberg, M.E.; Hogan, S.P. The eosinophil. Annu Rev Immunol 2006, 24, 147-174, doi:10.1146/annurev.immunol.24.021605.090720.
  3. Iriondo-DeHond, A.; Uranga, J.A.; Del Castillo, M.D.; Abalo, R. Effects of Coffee and Its Components on the Gastrointestinal Tract and the Brain-Gut Axis. Nutrients 2020, 13, doi:10.3390/nu13010088.
  4. Furuta, G.T.; Katzka, D.A. Eosinophilic Esophagitis. The New England journal of medicine 2015, 373, 1640-1648, doi:10.1056/NEJMra1502863.

It is very important to add a paragraph on the limitations of this study.

ANSWER: We thank the reviewer for this suggestion. We added the paragraph requested, as follows: “Our study has some limitations, tough. First, we did not have the possibility to dissect the effects of the single molecules of herbal extracts normally present in EDs, as their actual content and concentration are not disclosed. Second, our study is limited to short-term, acute effects, and did not explore the effects of long-term treatment with EDs. Finally, our study does not provide information concerning the functional consequences of eosinophilic infiltration on GI tract. Thus, further studies are needed to determine if and how the eosinophilic infiltration we observe could, when prolonged over time, contribute to EGIDs.”

Round 2

Reviewer 3 Report

Dear authors, 

Thank you for responding to all my comments. I hope they have been helpful.